

# Complete genome sequence and comparative genomics of the golden pompano (*Trachinotus ovatus*) pathogen, V*ibrio harveyi* strain QT520

Zhigang Tu[1,2,3], Hongyue Li[4], Xiang Zhang[1,4], Yun Sun[1,2] and Yongcan Zhou[1,4]

[1] State Key Laboratory of Marine Resource Utilization in South China Sea, Hainan University, P.R. China
[2] Key Laboratory of Tropical Biological Resources of Ministry of Education, Hainan University, Haikou, Hainan, China
[3] Hainan Academy of Ocean and Fisheries Sciences, Haikou, Hainan, China
[4] Hainan Provincial Key Laboratory for Tropical Hydrobiology and Biotechnology, College of Marine Science, Hainan University, Haikou, Hainan, China

Corresponding authors
Yun Sun, syshui207@126.com
Yongcan Zhou, zychnu@163.com

## ABSTRACT

*Vibrio harveyi* is a Gram-negative, halophilic bacterium that is an opportunistic pathogen of commercially farmed marine vertebrate species. To understand the pathogenicity of this species, the genome of *V. harveyi* QT520 was analyzed and compared to that of other strains. The results showed the genome of QT520 has two unique circular chromosomes and three endogenous plasmids, totaling 6,070,846 bp with a 45% GC content, 5,701 predicted ORFs, 134 tRNAs and 37 rRNAs. Common virulence factors, including ACF, IlpA, OmpU, Flagellin, Cya, Hemolysin and MARTX, were detected in the genome, which are likely responsible for the virulence of QT520. The results of genomes comparisons with strains ATCC 33843 (392 (MAV)) and ATCC 43516 showed that greater numbers genes associated with types I, II, III, IV and VI secretion systems were detected in QT520 than in other strains, suggesting that QT520 is a highly virulent strain. In addition, three plasmids were only observed in the complete genome sequence of strain QT520. In plasmid p1 of QT520, specific virulence factors (*cyaB*, *hlyB* and *rtxA*) were identified, suggesting that the pathogenicity of this strain is plasmid-associated. Phylogenetic analysis of 12 complete *Vibrio* sp. genomes using ANI values, core genes and MLST revealed that QT520 was most closely related to ATCC 33843 (392 (MAV)) and ATCC 43516, suggesting that QT520 belongs to the species *V. harveyi*. This report is the first to describe the complete genome sequence of a *V. harveyi* strain isolated from an outbreak in a fish species in China. In addition, to the best of our knowledge, this report is the first to compare the *V. harveyi* genomes of several strains. The results of this study will expand our understanding of the genome, genetic characteristics, and virulence factors of *V. harveyi,* setting the stage for studies of pathogenesis, diagnostics, and disease prevention.

## INTRODUCTION

*Vibrio harveyi* (*V. harveyi*) is a Gram-negative, halophilic bacterium that is recognized as an opportunistic pathogen of many commercially farmed marine invertebrate and vertebrate species (*Austin & Zhang, 2006*; *Cano-Gomez et al., 2011*). *V. harveyi* is capable of causing mass mortalities in aquaculture settings and has a major impact on the industry. Species affected throughout the world include the gilthead sea bream (*Sparus aurata*), European sea bass (*Dicentrarchus labrax*), common dentex (*Dentex dentex*), Senegalese sole (*Solea senegalensis*) and prawn (*Penaeus monodon*) (*Austin & Zhang, 2006*; *Chabrillón et al., 2005*; *Lavilla-Pitogo et al., 1990*; *Pujalte et al., 2003*; *Won & Park, 2008*; *Zorrilla et al., 2003*). *V. harveyi* was also reported to infect many aquaculture species in China and is now considered as one of the major pathogens in the fisheries industry (*Cui et al., 2014*; *Chen et al., 2004*; *Li & Xu, 1998*; *Zhang et al., 2010*).

Many studies have reported on the pathogenic mechanisms of *V. harveyi*, and variability in virulence mechanisms were found to be strain dependent toward different host species (*Austin & Zhang, 2006*; *Bai et al., 2008*; *Zhang & Austin, 2000*). Some reports have shown that extracellular products (ECPs) including proteases, phospholipases, hemolysins and cytotoxins, play significant roles in the pathogenicity of *V. harveyi* (*Liu & Lee, 1999*; *Liuxy, Lee & Chen, 1996*; *Zhang & Austin, 2000*). However, in fatal cases in *Artemia franciscana* nauplii and some cultured marine fishes, the hemolytic activity of *V. harveyi* was not a significant factor (*Soto-Rodriguez et al., 2003*; *Won & Park, 2008*). Siderophore production has been strongly correlated with fish mortality, and it is considered to be an essential virulence factor in some bacterial species (*Ratledge & Dover, 2003*), but the effect is less obvious in others (*Pedersen et al., 1997*). In recent years, Type III secretion systems (TTSS), bacteriophages, and quorum-sensing mechanisms, which play important roles in effector translocation, conversion of virulence factors, and regulation of virulence gene expression, respectively, have all been shown to be associated with virulence (*Henke & Bassler, 2004*; *Natrah et al., 2011*; *Ruwandeepika et al., 2012*). To further understand the pathogenesis and the associated virulence factors of *V. harveyi*, the determination of the genomic information of various *V. harveyi* strains is crucial.

In this study, we isolated a *V. harveyi* from a diseased golden pompano (*Trachinotus ovatus*) cultured in a deep sea cage in Qiaotou, China and named it QT520. Genome and comparative genome analyses were used in the current study to analyze the pathogenesis of *V. harveyi*. The results will help to characterize pathogenesis, diagnostics, and disease prevention.

## MATERIALS AND METHODS

### Ethics statement

All protocols for experiments involving live animals conducted in this study were approved by the Animal Experimental Inspection of Laboratory Animal Centre, Hainan University (Haikou, China), and the approval number is 20160709. We conscientiously abided by the ethical principles of animal welfare and the rule of laboratory animal center.

## Bacterial strain isolation

*V. harveyi* QT520 was initially isolated in Qiaotou, China from *T. ovatus* showing clinical signs of body rot. To obtain a pure culture, a single colony of strain QT520 was selected and transferred to fresh TCBS medium at least three times. Then, the clone was cultivated at 30 °C in 2216E liquid medium for 16 h. The cell morphology of QT520 and contamination of the culture with other microorganisms were assessed by light microscopy (BA410; Motic, Xiamen, China). The strain was archived at Hainan University and Hainan Academy of Ocean and Fisheries Sciences under accession number QT520.

## Genomic DNA extraction and whole genome sequencing

Genomic DNA was extracted using a Rapid Bacterial Genomic DNA Isolation Kit (Sangon Biotech, Shanghai, China) according to the manufacturer's instructions. The concentration and purity of the extracted DNA were determined using a DNA Qubit 2.0 (Invitrogen, Carlsbad, CA, USA). The extracted genomic DNA was used to construct next-generation libraries with an insert size of 500 bp using a NEBNext® Ultra™ DNA Library Prep Kit for Illumina® and PacBio RS II libraries with an insert size of 10 kb.

Genome sequencing was conducted by Sangon Biotech Co., Ltd. (Shanghai, China) using the next-generation Illumina MiSeq (Illumina, San Diego, CA, USA) and the third-generation PacBio RS II sequencing platform (Pacific Biosciences, Menlo Park, CA, USA), respectively. The raw sequences obtained from the third-generation PacBio RS II sequencing platform were assembled using Canu (*Koren et al., 2016*). Gapcloser and GapFiller were used to close the gaps with next-generation sequence reads where possible after assembly (*Boetzer & Pirovano, 2012*) and the sequence variants were detected and assembled by PrInSeS-G to obtain high-quality, whole-genome sequences (*Massouras et al., 2010*).

## Genome analysis

Prokka was used to predict and annotate prokaryotic ORFs, tRNAs and rRNAs (*Seemann, 2014*). Clusters of Orthologous Groups (COG) analysis was performed using RPS BLAST. Antibiotic resistance genes and genes encoding virulence factors were identified through BLAST searches of the Comprehensive Antibiotic Resistance Database (CARD) and the Virulence Factors of Pathogenic Bacteria Database (VFDB), respectively. Circular genome maps were generated using the CGView Server (*Grant & Stothard, 2008*) based on the information generated by the genome annotation.

## Phylogenetic analyses

Four phylogenetic trees were constructed based on average nucleotide identity (ANI), core genes, multilocus sequence typing (MLST), and the 16S rRNA gene, respectively. The ANI values were calculated using Jspecies (*Goris et al., 2007*) and phylogenetic tree analysis of 12 available complete genome sequences of *Vibrio* sp. strains (Table S1) based on ANI values was performed using the R package Ape with neighbor-joining methods (*Paradis, Claude & Strimmer, 2004*). The core genes of the 12 genomes were obtained using cd-hit 4.6.1, the corresponding protein sequences were aligned using MUSCLE 3.8.31 and the phylogenetic trees were generated using Treebest 1.9.2 (*Caputo et al., 2015*; *Mbengue et al., 2016*). Four

different housekeeping genes were used to construct the MLST phylogenetic tree: *toxR*, *vhhA*, *ompK* and *hsp60*. The 16S rRNA gene sequences and the corresponding amino acid sequences inferred from the four genes for MLST were respectively aligned with those of the most closely related species using the multiple alignment program ClustalW. Phylogenetic relationships between strain QT520 and closely related species were determined using the MEGA6 software. Phylogenetic trees were generated using the neighbor-joining (NJ) method with 1,000 randomly selected bootstrap replicates.

## Comparative genomics

Genomes of ATCC 33843 (392 (MAV)), ATCC 43516, and QT520 were compared through Mauve using the default parameters (*Darling et al., 2004*; *Mbengue et al., 2016*). The locations of different genes in the gene clusters were visualized using SVG 1.1, and ImageMagick 6.5.4–7 was used to convert SVG format to PNG.

## Determination of Median Lethal Dose (LD50)

The isolated QT520 strain was inoculated into 2216E liquid medium and incubated on a shaker (180 rpm) at 30 °C overnight, after which a culture containing $1.47 \times 10^8$ CFU/mL obtained. During challenge, the bacterial suspension was serially diluted 10-fold using Stroke-Physiological Saline Solution and was used to infect *T. ovatus* (10 fish/group) by intraperitoneal injection with $1.47 \times 10^8$ CFU/mL, $1.47 \times 10^7$ CFU/mL, $1.47 \times 10^6$ CFU/mL or $1.47 \times 10^5$ CFU/mL. The control group was injected with Stroke-Physiological Saline Solution. Fish deaths were recorded for 2 weeks and the Reed-Muench method was used to calculate the median lethal dosage.

## RESULTS

### General features of *V. harveyi* QT520

Through the use of a combination of sequencing approaches using the MiSeq and PacBio platforms, we were able to assemble five contigs with 227.76× coverage. The genome sequence of strain QT520 was deposited in the DDBJ/EMBL/GenBank under the accession numbers CP018680–CP018684. The project information, according to the minimum information about a genome sequence (MIGS) recommendation (*Field et al., 2008*), is shown in Table S2. The strain QT520, which was isolated from cultured *T. ovatus* exhibiting clinical signs of body rot, was Gram-negative and aerobic. After negative staining it was observed to be an oval, rod-shaped cell approximately 1.6–2.2 μm in length and 1.0–1.1 μm in width with a single polar flagellum (Fig. 1).

### Genomic information

The QT520 genome was a total of 6070846 bp, consisting of two circular chromosomes and three circular plasmids (chromosome I, 3,560,044 bp; chromosome II, 2,260,627 bp; plasmid p1, 113,574 bp; plasmid p2, 76,744 bp; plasmid p3, 59,857 bp) (Fig. 2). The genome had an overall G + C content of 45% (Table 1). A total of 3,377 open reading frames (ORFs) were detected in Chromosome I, including 3,195 coding sequences (CDSs), 118 tRNA genes, 34 rRNA genes, 29 miscRNA and 1 tmRNA; 2,425 CDSs (75.90% of

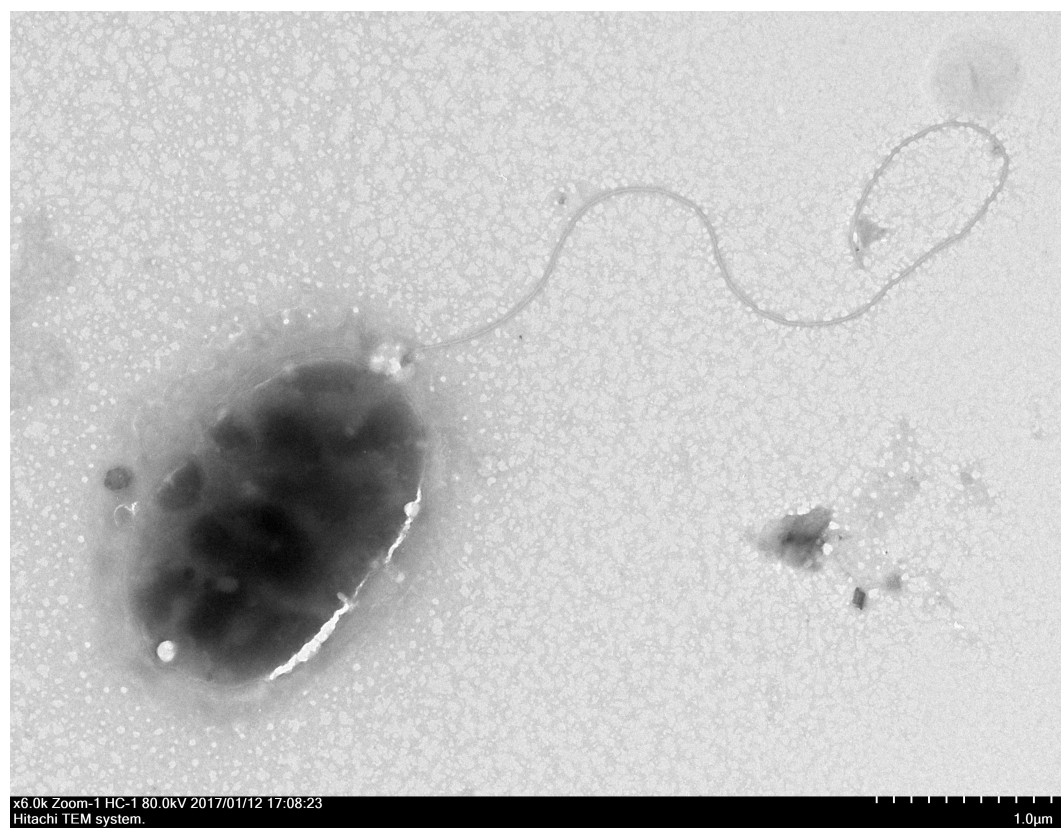

**Figure 1** **The electron micrograph of bacterium QT520 (X6000).**

3,195 CDSs) were annotated as functional genes, and 770 CDSs (24.10%) were annotated as hypothetical or uncharacterized genes. Chromosome II contained 2,061 ORFs, 2,030 CDSs, 16 tRNA genes, three rRNA genes and 12 miscRNA; 1,476 CDSs (72.71%) were annotated as functional genes, and 554 CDSs (27.29%) were annotated as hypothetical or uncharacterized genes. The chromosome maps (Fig. 3) display 3,657 ORFs (64.15% of the total number of predicted ORFs) that encode known functional proteins, and 423 ORFs having no known function in our COG functional categorization. Among the functionally predicted ORFs, 2,593 ORFs (63.55% of the COG-assigned ORFs) belonged to nine major COG functional categories: 527 ORFs in category R (General function prediction only), 329 ORFs in category K (Transcription), 287 ORFs in category E (Amino acid transport and metabolism), 276 ORFs in category T (Signal transduction mechanisms), 254 ORFs in category G (Carbohydrate transport and metabolism), 241 ORFs in category C (Energy production and conversion), 243 ORFs in category P (Inorganic ion transport and metabolism), 236 ORFs in category M (Cell wall/membrane/envelope biogenesis) and 200 ORFs in category J (Translation, ribosomal structure and biogenesis).

## Virulence factors of *V. harveyi* QT520

Virulence factors of strain QT520 are listed in Table S3. The results of the QT520 genome analysis revealed a number of common virulence determinants, such as ACF, IlpA,

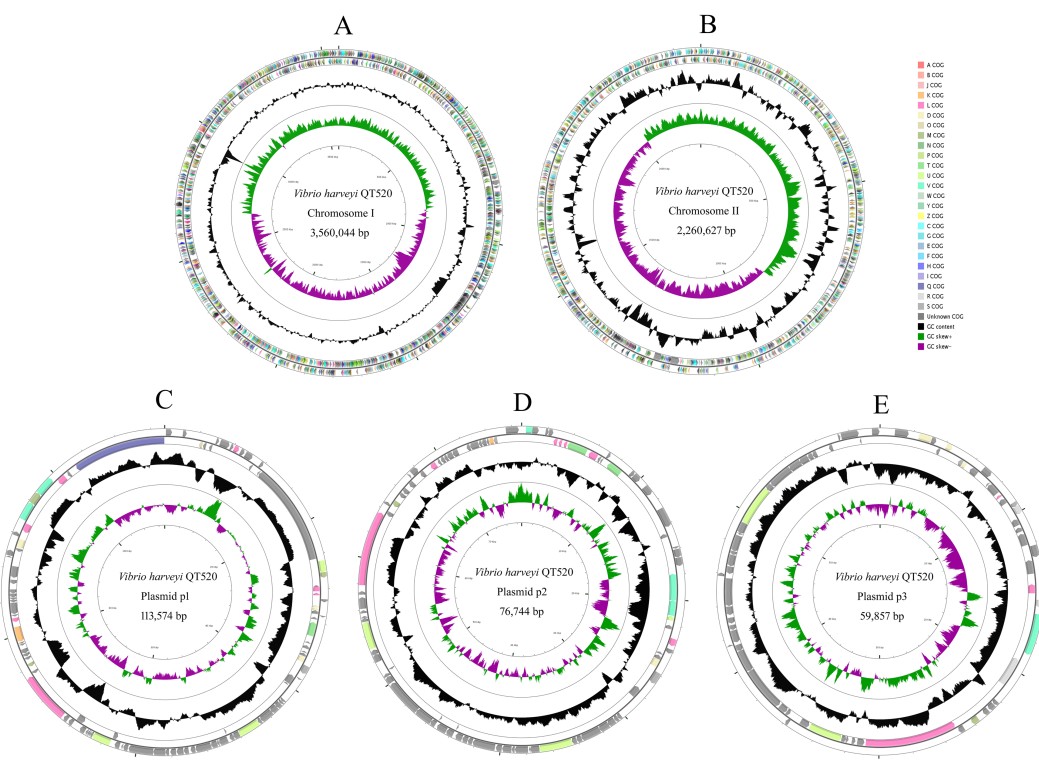

**Figure 2** **Genome map of *V. harveyi* QT520.** Chromosome I (A), Chromosome II (B), Plasmid p1 (C), p2 (D) and p3 (E). The outer circle indicates the location of all annotated ORFs, and all of them are colored differently according to the COG assignments. The middle circle with black peaks indicates GC content. The inner circle indicates GC-skew (green: GC-skew+; purple: GC-skew-).

**Table 1** **Comparison of the chromosomal properties of *V. harveyi* QT520, ATCC 33843 (392 (MAV)) and ATCC 43516.**

| Strain | QT520 | ATCC 33843 (392 (MAV)) | ATCC 43516 |
|---|---|---|---|
| Genome size(bp) | 6,070,846 | 5,881,490 | 6,038,881 |
| GC content | 45% | 44.96% | 44.90% |
| Open reading frames | 5,701 | 5,393 | 5,479 |
| Average length(bp) | 925 | – | – |
| % of ecoded gene | 96.19 | 95.18 | 95.80 |
| Annotated genes | 4,080 | – | – |
| Hypothetical proteins | 1,497 | 825 | – |
| tRNA | 134 | 131 | 133 |
| rRNA | 37 | 38 | 37 |
| Average nucleotide identity | 100.00 | 98.33 | 98.44 |
| GeneBank Accession No. | CP018680, CP018681 CP018682, CP018683 CP018684 | CP009467.2, CP009468.1 | CP014038.1 CP014039.1 |

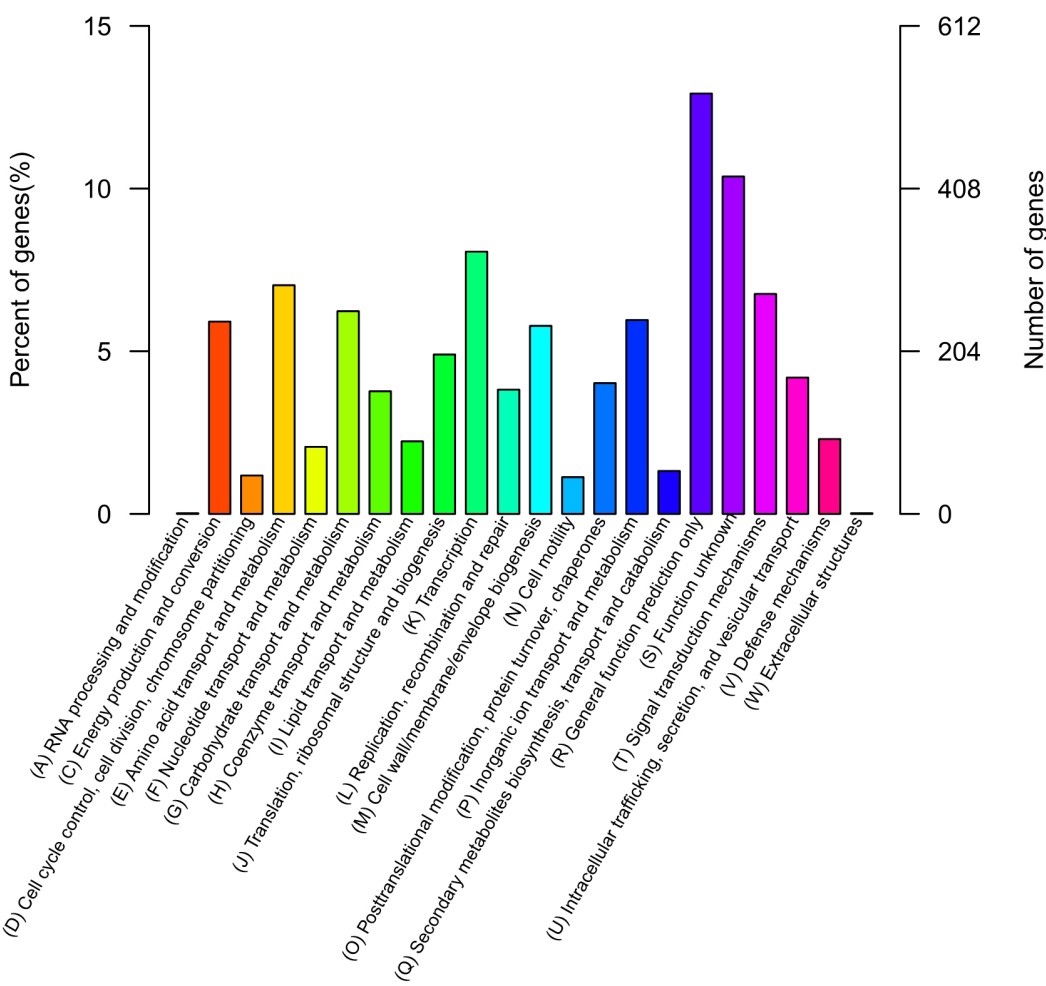

**Figure 3** **COG databases.** Functional categorization of all predicted ORFs in the genome of QT520 based on COG databases.

OmpU, Flagella, T6SS, TTSS, Cya, Hemolysin and MARTX. Factors responsible for iron acquisition, including IutA, Chu, Enterobactin, FeoAB and Mycobactin were also present. In addition, some adhesion proteins (Capsule/CapsuleI, Los, IlpA, MAM7 and OmpU), antiphagocytosis proteins (Alginate), efflux pumps, endotoxins, and stress proteins were also found in strain QT520.

## Phylogenetic analysis

Four comparative phylogenetic tree analysis methods were used to identify closely related strains of QT520: ANI, core genes, MLST, and 16S rRNA gene sequencing. For phylogenetic tree analysis using ANI, 11 complete genome sequences available in the NCBI GenBank database, as well as that of QT520, were obtained and the ANI values were calculated (Table S4). A phylogenetic tree based on ANI values was constructed (Fig. 4A). The tree identified a subgroup that contained 1114GL, LMB29, ATCC BAA-1116, ATCC 33843 (392 (MAV)), ATCC 43516 and QT520, which was designated Group I. Within this group,

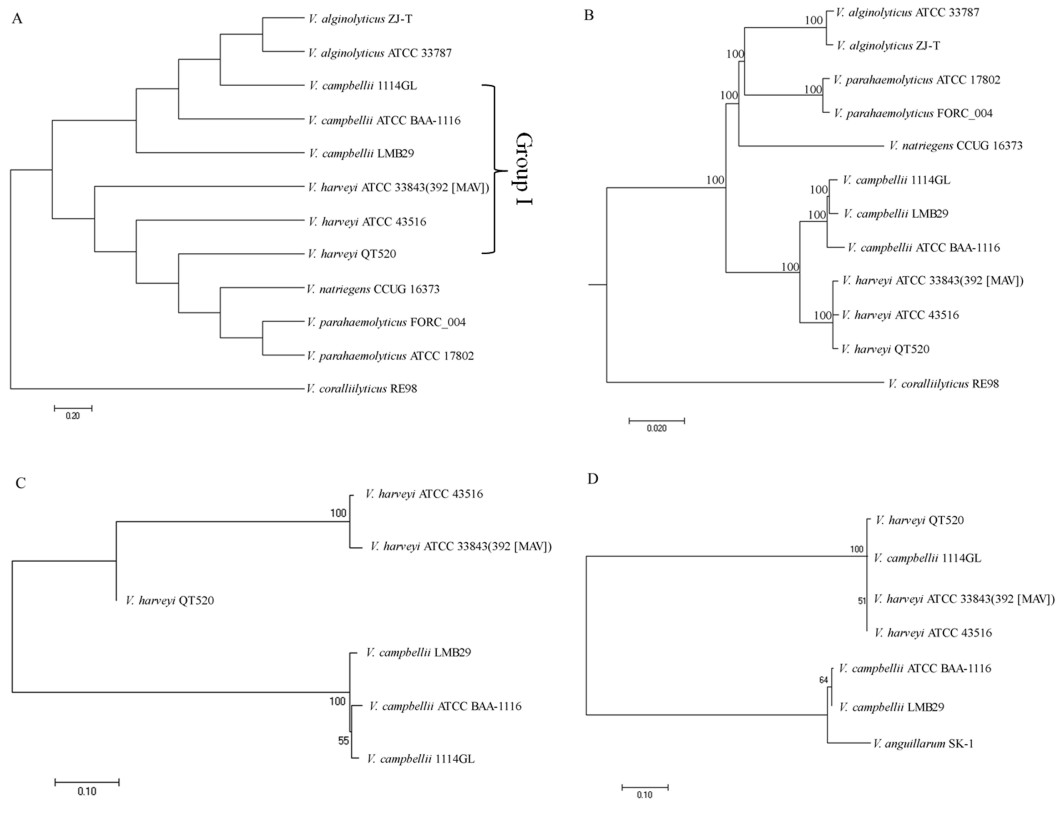

**Figure 4** **Phylogenetic tree analysis.** Phylogenetic tree analysis of 12 complete genome sequence of *Vibrio sp.* using ANI values (A) and core genes method (B). Phylogenetic tree analysis of Group I strains using MLST method (C) and 16S rRNA sequence method (D).

the genome of QT520 was most closely related to those of ATCC 33843 (392 (MAV)) and ATCC 43516 (98.49 and 98.59% of ANI values, respectively).

An additional phylogenetic tree of the 12 complete genome sequences described above was constructed using the core genes method (Fig. 4B). The results showed that ATCC 33843 (392 (MAV)) and ATCC 43516 were the closest in Group I and that QT520 was closely related to both strains, supporting that the ANI and core genes methods could distinguish them even at the strain level.

Another phylogenetic tree was generated using MLST with four different housekeeping genes: *toxR*, *vhhA*, *ompK* and *hsp60*. The DNA sequences of the four MLST genes in the Group I strains were compared and aligned to generate a phylogenetic tree (Fig. 4C). The results showed that the genomes of QT520, ATCC 33843 (392 (MAV)) and ATCC 43516 were again the closest relatives and those of 1114GL, LMB29 and ATCCBAA-1116 were the closest in Group I. However, comparative phylogenetic tree analysis using 16S rRNA sequences of the Group I strains revealed that strains QT520, 1114GL, ATCC 33843 392 MAV and ATCC 43516 were the closest relatives, being classified into one phylogenetic branch (Fig. 4D).

## Comparative genomics of *V. harveyi* ATCC 3843 (392 (MAV)) and ATCC 43516

Comparative genomic analyses were conducted to compare *V. harveyi* QT520 with its most closely related strains, *V. harveyi* ATCC 3843 (392 (MAV)) and ATCC 43516 (Fig. 5A). The results showed that the number of core genes among the three strains is 4,463. The number of specific genes in QT520, ATCC 3843, and ATCC 43516 are 458, 277 and 448, respectively.

Two genes (APP0488.1 and APP06774.1) identified as core genes encode a multi-antimicrobial extrusion efflux family protein and a xanthine-guanine phosphoribosyltransferase, respectively. The results suggested that the three *V. harvey i* strains may have multiple antibiotic resistances to tigecycline, tetracycline, streptomycin, kanamycin, ciprofloxacin and norfloxacin (*Kim et al., 2003*; *Kuroda & Tsuchiya, 2009*; *Teo, Tan & Poh, 2002*).

Virulence factor genes, including APP05600.1, APP06379.1, APP06380.1, APP06382.1, APP06383.1, APP06387.1, APP06388.1, APP07902.1, APP09130.1, APP09132.1, APP09134.1, APP09207.1 and APP09259.1 were only observed in strain QT520 based on the comparative analysis results (Table S3 and Fig. 5A). These specific genes encode Type IV pili, O-antigen, LPS, Capsule, Cya, hemolysin, MARTX and TTSS (SPI-1), respectively. These genes are well known to play important roles in attachment, expression of other virulence factors, preventing phagocytosis, anti-inflammatory effects, hemolytic activities and a cytotoxicity (*Ruwandeepika et al., 2012*; *Zhang & Austin, 2000*), suggesting that they may be responsible for the virulence associated with the QT520 strain.

Genes associated with type I secretion systems (T1SS) and type VI secretion systems (T6SS) were detected in QT520, but not in ATCC 33843 (392 (MAV)) and ATCC 43516. The numbers of type II, III and IV secretion systems (T2SS, T3SS, and T4SS) related genes in QT520 was 28, 32, and 21, respectively; the numbers of T2SS, T3SS, and T4SS related genes in ATCC 33843 (392 (MAV)) was 7, 12, and 4, respectively. The numbers of genes T2SS, T3SS, and T4SS related genes in ATCC 43516 was 7, 14, and 4, respectively. Greater numbers of genes associated with the T1SS, T2SS, T3SS, T4SS and T6SS were detected in the genome sequence of QT520 than in strains ATCC 33843 (392 (MAV)) and ATCC 43516 (Table 2 and Fig. 5). In addition, T1SS were detected in regions of chromosome I (APP04465.1), chromosome II (APP08059.1 and APP08730.1) and plasmid p1 (APP09130.1 and APP09132.1) of QT520. T2SS and T3SS were detected in the regions APP04623.1–APP08089.1 and APP04087.1–APP08428.1 in chromosomes I and II, respectively. Nine T4SS-encoding genes were detected in the insertion region in QT520 compared to ATCC 3843(392(MAV)) and ATCC 43516 (Fig. 5B and Fig. 5C), and other T4SS-encoding genes were primarily distributed in regions of chromosome I (APP05190.1–APP05200.1) and chromosome II (APP07457.1 to APP08116.1). T6SS-related genes were detected in regions of chromosome I (APP04433.1 to APP04446.1) and chromosome II (APP07649.1 to APP07972.1).

Three plasmids were detected in *V. harveyi* strain QT520 that were not observed in strains ATCC 3843 (392 (MAV)) and ATCC 43516. In particular, three important virulence factor genes (*cyaB*, *hlyB* and *rtxA*) were present in plasmid p1.

Peer]

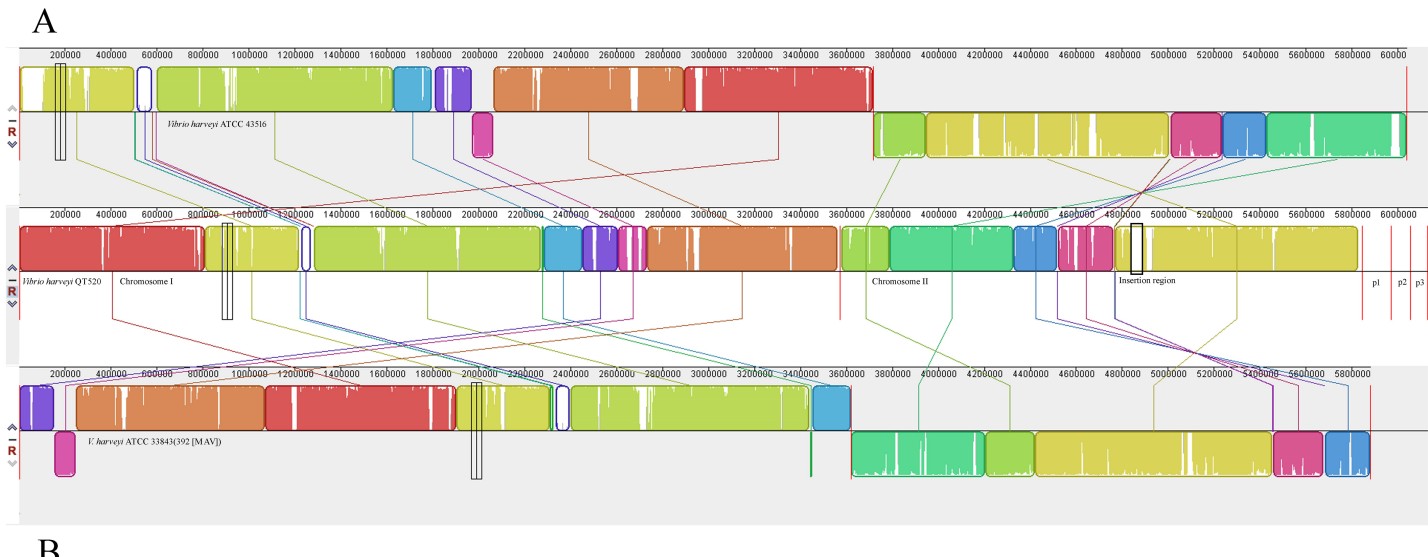

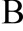

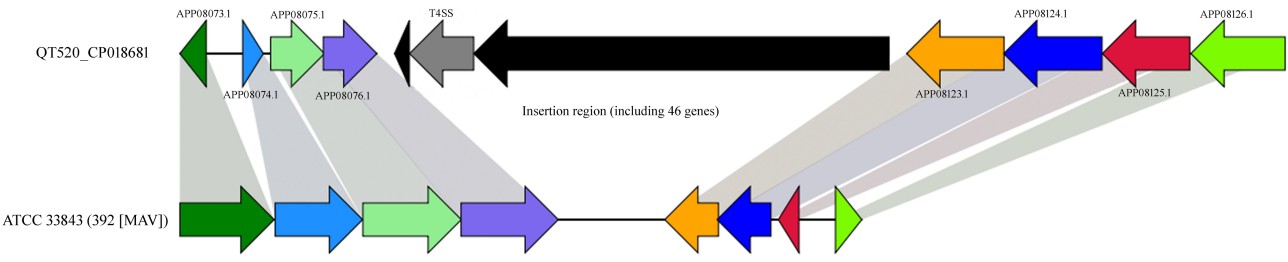

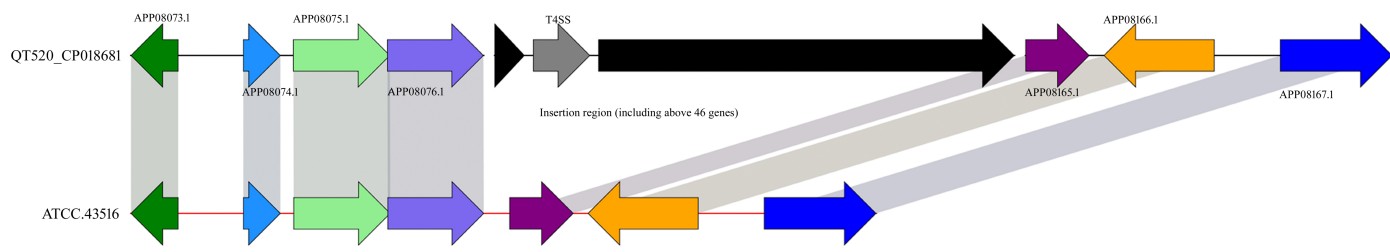

**Figure 5** **Mauve alignment of the genome.** *Vibrio harveyi* QT520 and genomes of ATCC 33843 (392 (MAV)) and ATCC 43516 (A). The insertion region of QT520 compared with ATCC 33843 (392 (MAV)) (B). The insertion region of QT520 compared with ATCC 43516 (C). The collinear blocks of the same color represent the highly homologous regions. The genomes were drawn to scale based on the strain QT520 genome, showing one common insertion region in chromosome II of strain QT520 compared to genomes of ATCC 33843 (392 (MAV)) and ATCC 43516.

**Table 2   The quantity of TnSS gene in the genome sequences of QT520, ATCC 33843 (392 (MAV)) and ATCC 43516.**

| Strain | TnSS | | | | |
|---|---|---|---|---|---|
| | I | II | III | IV | VI |
| QT520 | 5 | 28 | 32 | 21 | 45 |
| ATCC 33843 (392 (MAV)) | 0 | 7 | 12 | 4 | 0 |
| ATCC 43516 | 0 | 7 | 14 | 4 | 0 |

**Table 3   Artificial infection results of the strain QT520.**

| Group | Concentration of QT520 (CFU/mL) | Number | Accumulative death number | | | | | | | Total deaths | Mortality (%) |
|---|---|---|---|---|---|---|---|---|---|---|---|
| | | | 1d | 2d | 3d | 4d | 5d | 6d | 7d | | |
| 1 | $1.47 \times 10^8$ | 10 | 9 | 1 | 0 | 0 | 0 | 0 | 0 | 10 | 100 |
| 2 | $1.47 \times 10^7$ | 10 | 6 | 2 | 0 | 0 | 0 | 0 | 0 | 8 | 80 |
| 3 | $1.47 \times 10^6$ | 10 | 0 | 4 | 0 | 0 | 0 | 0 | 0 | 4 | 40 |
| 4 | $1.47 \times 10^5$ | 10 | 0 | 0 | 0 | 0 | 0 | 0 | 0 | 0 | 0 |
| Control group | 0.8%NaCl | 10 | 0 | 0 | 0 | 0 | 0 | 0 | 0 | 0 | 0 |

## LD$_{50}$

Artificial infection results showed that the LD$_{50}$ of QT520 was $2.5 \times 10^5$ bacteria per fish, suggesting that this bacterium is strongly pathogenicity to *T. ovatus* (Table 3).

## DISCUSSION

Since species within the Harveyi clade have a high degree of genetic and phenotypic similarity (*Sawabe, Kitatsukamoto & Thompson, 2007*), identification of *V. harveyi* strains is a challenging task. For example, the species *V. harveyi*, *V. campbellii* and *V. rotiferianus* share approximately 99% sequence identity within the 16S rRNA gene (*Gomez-Gil, 2003*). These similarities have confounded typing schemes and resulted in documented misidentifications (*Gauger & Gómez-Chiarri, 2002*; *Gomez-Gil et al., 2004*). Recently, novel phylogenetic analysis approaches have been suggested, specifically, ANI, core genes and MLST. The ANI and core genes method was performed using the 12 complete genome sequences available for *Vibrio* sp. In this study, the analysis based on core genes and MLST revealed that QT520, ATCC 3843 (392 (MAV)) and ATCC 43516 were the most closely related and that 1114GL, LMB29 and ATCCBAA-1116 were the closest in Group I, which was consistent with the analysis using ANI values. However, these strains could not be differentiated using the 16S rRNA phylogenetic tree, suggesting that ANI, core genes and MLST methods can provide a greater level of resolution between *V. harveyi* and *V. campbellii* compared to the 16S rRNA method.

The pathogenicity of *V. harveyi* strains is related to a number of factors, including secretion of ECPs (containing substances such as proteases, hemolysins, and lipases) (*Teo, Zhang & Poh, 2003*; *Zhang & Austin, 2000*), a bacteriocin-like substance (*Prasad et al., 2005*), quorum sensing capabilities (*Henke & Bassler, 2004*), susceptibility to bacteriophage

infection (*Oakey & Owens, 2000*) and siderophore production (*Owens, Austin & Austin, 1996*). The observed $LD_{50}$ of QT520 was $2.5 \times 10^5$ bacteria per fish, suggesting that this bacterium is a highly virulent strain towards *T. ovatus*. Various virulence factor genes, including ACF, IlpA, MAM7, OmpU, Type IV pili, Flagellin, Cya, Hemolysin and MARTX, were observed in this strain. These genes may be responsible for the high virulence associated with the QT520 strain.

The ability of pathogens to obtain iron from their host is central to their survival (*Ratledge & Dover, 2003*). The genome of the QT520 strain encodes the aerobactin siderophore receptor (detected at the APP07707.1 region on chromosome II), heme receptors (at the region APP04364.1 on chromosome I and the region APP07339.1 on chromosome II), and iron ABC transport (APP03856.1–APP03858.1 and APP03907.1 on chromosome I). These iron uptake related-genes may play key roles in the survival of the QT520 strain in host cells.

Secretion systems can transport various virulence factors outside of the bacterial cell and allow bacteria to communicate within the environment in which they live. Greater numbers of genes encoding T1SS, T2SS, T3SS and T4SS and T6SS components were found in strain QT520 than in strains ATCC 33843 (392 (MAV)) and ATCC 43516. Interestingly, the T1SS and T6SS were only found in QT520. The T1SS is employed for secreting proteins, includes many adhesins, proteases, and toxins that are delivered into host cells, and can secrete hemolysin in the *E. coli* (*Dalbey & Kuhn, 2012*). The T2SS is required for secretion of exotoxins, including the cholera toxin (Ctx) (*Abendroth, Kreger & Hol, 2009*). The T3SS, which is found in various pathogenic Gram-negative bacterial genera, such as *Salmonella*, *Shigella*, *Yersinia*, *Pseudomonas*, and enteropathogenic *Escherichia coli* (EPEC) (*Galan & Wolf-Watz, 2006*), serve several well defined functions in pathogenesis. The T4SS is responsible for the transport of virulent proteins or DNA into eukaryotic cells as well as for the conjugative transfer of plasmids from one bacterium to another (*Fronzes, Christie & Waksman, 2009*) and were found in the insertion regions, suggesting that the T4SS may play important roles in the pathogenicity of QT520. The T6SS, which is associated with cytotoxic effectors (*Costa et al., 2015*; *Unterweger et al., 2014*), translocates toxic effector proteins into eukaryotic and prokaryotic cells and has a pivotal role in pathogenesis and bacterial competition (*Ho, Dong & Mekalanos, 2014*; *Zoued et al., 2014*), and this system was also reported in the pathogenic *Vibrio harveyi* strains ZJ0603 and CAIM 1792 (*Huang et al., 2012*; *Espinoza-Valles et al., 2012*). This secretion system functions as a group of toxin proteins by transporting various bacterial effectors into eukaryotic cells, resulting in host cell death (*Costa et al., 2015*), and its presence suggests that QT520 is a highly virulent strain based on our genome analysis. However, further analysis of the mechanisms of these secretion systems in *V. harveyi* is required.

Interestingly, we observed three plasmids in the complete genome sequence of strain QT520, which have not been identified in strains ATCC 33843 (392 (MAV)) and ATCC 43516 (*Wang et al., 2014*). Through comparative genomic analysis, three specific virulence factors (Cya, Hemolysin and RTX toxin), encoded by *cyaB*, *hlyB* and *rtxA*, were observed in plasmid p1 of strain QT520, suggesting that the pathogenicity of this strain is closely related to plasmid p1. Studies have also reported that a plasmid, pVCR1, was harbored by the highly virulent *V. harveyi* ORM4 strain, suggesting its involvement in the virulence

phenotype (*Schikorski et al., 2013*). The virulence of *V. harveyi* was believed to be acquired by association with genetically mobile elements, such as plasmids or transposons (*Austin & Zhang, 2006*). Non-virulent *V. harveyi* strains can become virulent after plasmid uptake or bacteriophage-mediated transfer of toxin gene(s) (*Oakey & Owens, 2000*). Thus, we inferred that the pathogenicity of QT520 has been acquired by the incorporation of plasmids.

Through comparative analyses of the complete genomes of the three *V. harveyi* strains, a majority of the genes analyzed were observed to be core genes, including the antibiotic resistance genes reported previously in the complete genome sequences of *V. harveyi* ATCC 3843 (392 (MAV)) and ATCC 43516. This suggests that the three *V. harveyi* strains likely possess similar antibiotic resistances to tigecycline, tetracycline, streptomycin, kanamycin, ciprofloxacin and norfloxacin. In addition, QT520 exhibited resistance against tetracycline, streptomycin and kanamycin.

## CONCLUSIONS

*V. harveyi*, an opportunistic pathogen of many maricultured animals, can cause mass mortalities in aquaculture species, posing a considerable threat to the industry. *V. harveyi* QT520, which was isolated from diseased deep sea cage-cultured golden pompano, was observed to be a virulent isolate with an LD50 of $2.5 \times 10^5$ bacteria per fish. To understand its pathogenesis, the genome of *V. harveyi* QT520 was sequenced. This genome was observed to consist of two circular chromosomes and three plasmids, totaling 6,070,846 bp with a 45% GC content, as well as containing 5,701 predicted ORFs, 134 tRNAs and 37 rRNAs.

Phylogenetic analysis of 12 complete genomes of *Vibrio* sp. using ANI values and core genes revealed that QT520 was most closely related to ATCC 33843 (392 (MAV)) and ATCC 43516, indicating that QT520 belongs to the species *V. harveyi*.

Common virulence factors, including ACF, IlpA, OmpU, Flagellin, Cya, Hemolysin and MARTX, were detected. These factors may be responsible for the virulence associated with colonization by QT520. Additionally, greater numbers of genes encoding types I, II, III, IV and VI secretion systems were detected in the genome of QT520 than in strains ATCC 33843 (392 (MAV)) and ATCC 43516, suggesting that strain QT520 has the capacity for a highly virulent phenotype. It is worth mentioning that three specific virulence factor genes (*cyaB*, *hlyB* and *rtxA*) were contained in plasmid p1 in this strain, suggesting that the pathogenicity of this strain is plasmid associated. Comparative genome analysis of QT520, ATCC 33843 (392 (MAV)) and ATCC 43516 revealed that the majority of ORFs were core genes, including two antibiotic encoding genes, suggesting these strains are resistant to multiple antibiotics.

### Funding

This research was supported financially by the National Natural Science Foundation of China (No. 31560725, No.31702379, No. 41666006), the National Marine Public Welfare Research Project of China (No. 201405020-4), the Key Research Project of Hainan Province

(ZDKJ2016011), the Natural Science Foundation of Hainan Province (No. 2016CXTD005, No. 20163054), and the Hainan University Campus Team Project 2017 (hdkytg201704). The funders had no role in study design, data collection and analysis, decision to publish, or preparation of the manuscript.

### Grant Disclosures

The following grant information was disclosed by the authors:
National Natural Science Foundation of China: 31560725, 31702379, 41666006.
National Marine Public Welfare Research Project of China: 201405020-4.
Key Research Project of Hainan Province: ZDKJ2016011.
Natural Science Foundation of Hainan Province: 2016CXTD005, 20163054.
Hainan University Campus Team Project 2017: hdkytg201704.

### Competing Interests

The authors declare there are no competing interests.

### Author Contributions

- Zhigang Tu performed the experiments, analyzed the data, contributed reagents/materials/analysis tools, wrote the paper, prepared figures and/or tables, reviewed drafts of the paper.
- Hongyue Li analyzed the data, contributed reagents/materials/analysis tools, prepared figures and/or tables.
- Xiang Zhang analyzed the data, contributed reagents/materials/analysis tools.
- Yun Sun analyzed the data, wrote the paper, prepared figures and/or tables, reviewed drafts of the paper.
- Yongcan Zhou conceived and designed the experiments, reviewed drafts of the paper.

### Animal Ethics

The following information was supplied relating to ethical approvals (i.e., approving body and any reference numbers):

All protocols for experiments involving live animals conducted in this study were approved by the Animal Experimental Inspection of Laboratory Animal Centre, Hainan University (Haikou, China), and the approval number is 20160709.

### DNA Deposition

The following information was supplied regarding the deposition of DNA sequences:
CP018680–CP018684.

### Data Availability

The raw data is included in the manuscript (Table 1–5, Figure 1–3, and Appendix A. Supplementary data).

### Supplemental Information

Supplemental information for this article can be found online at http://dx.doi.org/10.7717/peerj.4127#supplemental-information.

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
