# Peer review of "Complete genome sequence and comparative genomics of the golden pompano (Trachinotus ovatus) pathogen, Vibrio harveyi strain QT520"

_PeerJ, doi:10.7717/peerj.4127_

## Round 0.1 · original submission · Major Revisions

The manuscript has now been seen by two expert reviewers and based on their recommendations, a revised version of the article can be considered for publication, provided that all of the reviewer concerns can be addressed by the authors.

Reviewer 1 ·

Basic reporting

The authors describe the genome of a newly isolated strain (QT520) of Vibrio harveyi, and compare it with related Vibrio harveyi strains. The genome sequence of V. harveyi QT520 will be of interest to the scientific community and may lead to a better understanding of its pathogenicity. However, the manuscript is very poorly written; the quality of written English is unacceptable in several occasions and is not suitable for publication unless extensively edited.

* Several sections require language edits, including but not limited to:

- Line 29: “consists two circular chromosomes”
- Lines 34-35: “suggesting strain QT520 were provided with the capacity for a highly virulent phenotype”
- Line 41: “and first compared the genome with other V. harveyi strains”
- Line 152 “This bacterium was a Gram-negative, aerobic”
- Line 158 “Chromosome I was detected of 3211 ORFs”
- Line 161 “Chromosome II was contained of 2044 ORFs,”
- Line 326 "Strains QT520 were exhibited resistance to tetracycline and ciprofloxacin"

* Figure 2: Both A and B are labelled Chromosome I and should be changed

* Figure 4B legend missing genome name

* Table 1 is missing details of Illumina sequencing technology and related assembly methods

* Sequence data is available in public domain but the size of the chromosome and plasmids mentioned in the paper do not match with that in GenBank, examples including but not limited to:
- Line 156 - “chromosome I of 3571617 bp”; Chromosome 1 in GenBank (CP018680.2) is 3560044bp
- Line 156 - “plasmid p1 of 124998bp”; Plasmid 1 in GenBank (CP018682.2) is 113574bp

Experimental design

The Phylogenetic Analysis section lacks credibility and fails to provide an accurate picture with respect to the phylogenetic position of Vibrio harveyi QT520. Which single copy genes were used to generate Fig 4A? While no bootstrap values are provided for Figure 4A, bootstrap values of 24 and 43 in Fig. 4C are way too low to provide any meaningful information. The authors also used different sequence alignment, tree building method and a different set of organisms for the three phylogenetic trees, which are thus not easily comparable. In the single-copy gene tree, QT520 clusters with the other two Vibrio harveyi strains while in the 16S tree QT520 is closer to V. campbelli 1114GL than to other V. harveyi strains. Since this is the first report describing Vibrio harveyi QT520, the authors need to provide more conclusive evidence confirming isolate QT520 is indeed Vibrio harveyi. While the authors mention that 16S rRNA based trees failed to differentiate between the different strains, the statement needs to be substantiated with more robust tree building algorithm and/or complemented with whole genome average nucleotide identity (ANI) values with reference V. harveyi genomes.

Validity of the findings

* Provide more specifics on how the genomes were compared using Mummer. Include the run parameters used including cutoffs

* Line 208: Define “perfectly matched ORFs”

* Lines 210 - 213: Provide literature reference suggesting that the two core genes are responsible for multiple antibiotic resistance to substantiate the claim that QT520 possesses similar antibiotic resistance features as the other two V. harveyi strains

* Lines 218-221: Provide literature reference

Additional comments

* In the General features section, consider discussing the physiological/biochemical properties separately from the genomic information.

* In the LD50 section consider including some information comparing pathogenicity and LD50 values of ATCC 3843 and ATCC 43516 with QT520

Reviewer 2 ·

Basic reporting

The work from Tu et al. "Complete genome sequence and comparative genomics of the golden pompano (Trachinotus ovatus) pathogen, Vibrio harveyi strain QT520" is interesting work adding up to our understanding of the pathogenicity and the population genomics of Vibrio sp. Beyond doubt, it is an interesting work worth to be published. Nevertheless, several issues regarding the use of language have to be addressed and corrected prior to publication: e.g. L51-54 change to "... causing mass mortalities in aquaculture species having a major impact on the industry. Species affected throughout the world include the gilthead sea bream ... and prawn (Penaeus monodon)." L55-57 change to "...to infect many aquaculture species in China and is now considered as one of the major pathogens to the fisheries industry". This goes throughout the whole manuscript. I recommend a native speaker or a professional editor to go through the manuscript and correct syntax errors and misuse of English.

Experimental design

Line 98-100: Even though I understand what they mean, it is confusing. Please rephrase in a clear way.

Validity of the findings

Table 1 and table 3 can be moved to the supplement. Additionally, Table 1 "Sequencing_meth" was combined Illumina Miseq and PacBio; please correct accordingly.
Section Discussion Lines 250-256: Please remove completely. Does not adding anything to the scientific impact of the paper and delivers confusing messages. It is well known that different methods/techniques/approaches needed to answer different scientific questions. Moreover, even though it is clear that the QT520 strain is a pathogen isolated in China, the prevalence as a major aquaculture pathogen in China (or anywhere else in the world) is not addressed in this work and not supported by the data. Therefore, a reader should perceive it as another pathogenic strain of Vibrio harveyi, increasing our knowledge of molecular ecophysiology of V. harveyi.
Furthermore, it is highly recommended to the authors to submit the strain at least to one culture collection and make it available to the scientific community; I am confident several scientists would be interested to know how they can get access to the strain and this should be mentioned in the manuscript.

---

## Round 0.2 · Minor Revisions

Thank you for addressing the reviewer's comments. It seems that there are a couple of more minor comments, that can easily be fixed. As soon as you can address these, the manuscript should be accepted.

Reviewer 1 ·

Basic reporting

no comment

Experimental design

no comment

Validity of the findings

no comment

Additional comments

Line 71: Remove duplicate ‘a significant factor’

Lines 103 - 111 : The authors provide conflicting information about library prep and sequencing method. Library prep was done for Illumina and sequencing was performed by PacBio RS II? Again in the results section the authors mention that sequencing was performed using a combination of PacBio and MiSeq. Please clarify..

Lines 348-49 : Remove last sentence of the paragraph

---

## Round 0.3 · accepted · Accept

Thank you for addressing all the reviewer comments.